biochemistry/environmental science

anammox, carbon source, $COD/NO_2^-$-N ratios, nitrite, nitrogen removal performance

**Author for correspondence:**
Jun Li
e-mail: bjutjglijun@163.com

# Effect of organic matters on anammox coupled denitrification system: when nitrite was sufficient

Jingyue Yang, Jun Li, Zhaoming Zheng, Liangang Hou, Dongbo Liang, Yiqi Sun and Xiaoran Ma

Beijing University of Technology College of Architecture and Civil Engineering, Beijing, People's Republic of China

JY, 0000-0001-7217-4788

Anaerobic ammonia oxidation (anammox) and denitrification can work together to weaken the influence of organic matter on anaerobic ammonia oxidation bacteria (AAOB) and improve nitrogen removal performance. As the common substrate of anammox and denitrification, nitrite will also affect nitrogen removal performance when it is insufficient, which is not conducive to reflect the endurance of anammox reactor to organic matter. The UASB continuous flow experiment was carried out to investigate the effect of the concentration of glucose and sodium acetate on nitrogen removal performance of anammox reactor under the condition of sufficient nitrite. With glucose as the organic matter, when the chemical oxygen demand (COD) concentration increased to 200 mg l$^{-1}$, nitrogen removal performance of the system began to deteriorate significantly, and the anammox activity was significantly inhibited. With sodium acetate as the organic substance, the anammox activity was affected when the COD was 20 mg l$^{-1}$. Adequate nitrite could relieve the inhibition of the coupling system by a low concentration (COD < 200 mg l$^{-1}$) of glucose organic matter. However, it could not relieve the inhibitory effect of sodium acetate. With the increase of organic concentration, the biological density of AAOB in granular sludge gradually decreased, while the biological density of denitrifying bacteria increased gradually.

## 1. Introduction

The traditional biological nitrogen removal process adopts nitrification and denitrification technology, which has high aeration energy consumption and large sludge production, and requires the additional carbon source [1]. Anaerobic ammonia

oxidation (anammox) is an economical and convenient way to remove nitrogen. Anaerobic ammonia oxidation bacteria (AAOB) use $NH_4^+$-N as electron donor and $NO_2^-$-N as electron acceptor to produce $N_2$ and a small amount of $NO_3^-$-N [2]. However, the growth rate of AAOB is low, and its doubling time is up to 11 days [3]. The study found that organics can inhibit nitrogen removal performance of AAOB [3–5]. There are two main inhibiting mechanisms of organics: one is substrate competition-inhibition. In anammox reactor, non-toxic organics can accelerate the growth of denitrifying bacteria and denitrifying bacteria compete with AAOB for the common substrate nitrite [6,7]. Another is toxic inhibition. Toxic organics in reactor could directly inhibit the activity of AAOB [6,8,9].

Anammox coupled denitrification system could relieve the effect of organics on AAOB and achieve good nitrogen removal performance [10,11]. However, the types of carbon sources and the concentration of nitrite affect the performance of coupled system [12,13]. The more complex the metabolic pathway of organic matter is, the lower nitrogen removal rate of sludge will be [14,15]. Anammox granular sludge has a higher biomass density, and there is mass transfer resistance in granular sludge. The surface of granular sludge contains abundant extracellular polymeric substance (EPS), which is helpful to alleviate the inhibition of AAOB by environmental changes [16–18]. The study of Qin *et al.* [19] indicated that higher concentration of glucose in the reactor could significantly inhibit the anammox activity of the sludge and reduce the removal rate of ammonia nitrogen obviously; but when the substrate of nitrite was sufficient and the concentration of glucose was low, the inhibition of anammox activity was relieved. The researchers also investigated the effect of organics concentration on nitrogen removal performance of anammox reactor through long-term experiments [7,20,21]. However, in the majority of the related research [22,23], the nitrite in the reactor was in an insufficient state, and the lack of nitrite inhibited the activity of AAOB, which could not reflect the endurance of the anammox reactor to organic matter. Moreover, in existing studies [24,25], there was no systematic comparison of the effects of different organics on anammox coupled denitrification system when nitrite was sufficient.

In this study, to explore the influence threshold of organic matters on anammox under the condition of sufficient nitrite, and to analyse the competitive inhibition characteristics on substrates of anammox coupled denitrification system, the effect of glucose and sodium acetate concentration on nitrogen removal performance of anammox reactor was investigated in a continuous flow reactor. In the reactor, nitrite was maintained sufficient as substrate to avoid competitive inhibition. And the relationship between the parameters was explored through Pearson correlation analysis. High-throughput sequencing analysis was used to study the population changes of granular sludge and to explore the further impact of organics on AAOB.

# 2. Material and methods

## 2.1. Test device

Figure 1 shows the UASB (upflow anaerobic sludge blanket) anammox reactor, which was conducive to sludge sedimentation and granulation. The effective volume of the reactor was 10 l. And the black soft material was wrapped around the reactor to avoid light. The influent was driven into the bottom of the reactor by a peristaltic pump. The temperature in the reactor was controlled by a water bath, and the Kaldnes ring carriers with a diameter of 2.5 cm were filled above the granular sludge to reduce the loss of sludge.

## 2.2. Inoculated and medium

### 2.2.1. Inoculated sludge

Four litres of anammox granular sludge taken from a UASB anammox reactor which has been operating for 2 years in the laboratory was inoculated into the reactor, and the volatile suspended solids (VSS) was about 4500 mg l$^{-1}$. The average particle size of the granular sludge was 2 mm.

### 2.2.2. Synthetic wastewater

The synthetic contaminated water was fed to the reactor. $NH_4^+$-N and $NO_2^-$-N were supplemented to the medium in the form of $NH_4Cl$ and $NaNO_2$, respectively. The medium was made up of $NaHCO_3$ 1.25 g l$^{-1}$, $KH_2PO_4$ 0.01 g l$^{-1}$, $MgSO_4 \cdot 7H_2O$ 0.3 g l$^{-1}$ and $CaCl_2$ 0.0056 g l$^{-1}$. The addition of trace elements was referred to the literature [7,26]. And the experiment was divided into four stages while the hydraulic retention time (HRT) was maintained at 0.96 h to maintain a higher load and a larger hydraulic shearing

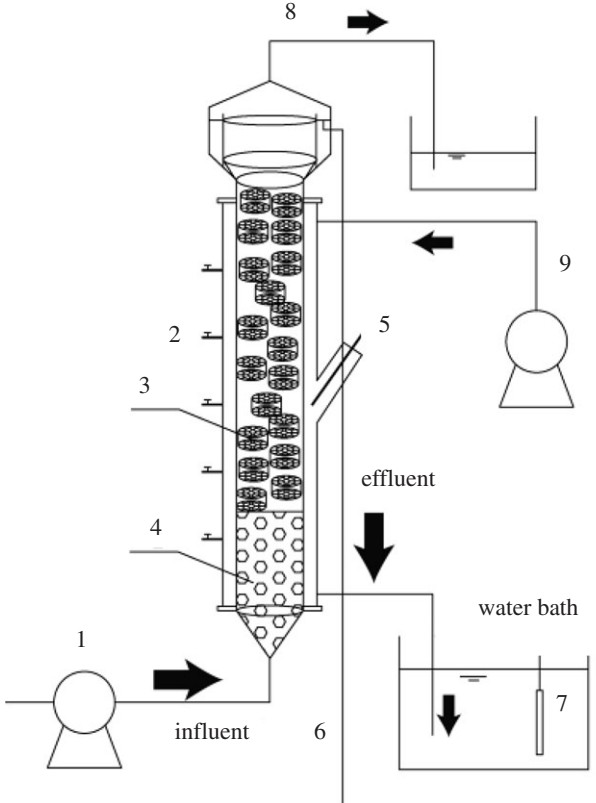

**Figure 1.** Schematic diagram of UASB reactor. 1, Influent pump; 2, sampling port; 3, Kaldnes ring; 4, granular sludge; 5, thermometer; 6, effluent; 7, heating rod; 8, outlet port; 9, water bath pump.

**Table 1.** The operation of continuous flow reactor at different stages with glucose as organic matter.

| stage | time (d) | concentration (mg l$^{-1}$) | | |
| | | $NH_4^+$-N | $NO_2^-$-N | COD (glucose) |
|---|---|---|---|---|
| 1 | 1–30 | 50 | 66 | 0 |
| 2 | 31–60 | 50 | 100 | 50 |
| 3 | 61–90 | 50 | 110 | 100 |
| 4 | 91–120 | 50 | 110 | 200 |

force in the reactor. The concentrations of NH$_4$Cl, NaNO$_2$ and organic matter (glucose and sodium acetate) in influent are shown in tables 1 and 2. To ensure the sufficient substrate of nitrite, the initial concentration of nitrite in the influent was the sum of nitrite theoretically consumed by denitrification (assuming that all the added chemical oxygen demand (COD) was consumed) and anammox. Then it increased correspondingly at different stages according to the experiment. The concentration of organic matter in the influent was expressed by COD$_{Gr}$, COD$_{Gr}$/glucose = 1.07 g g$^{-1}$, COD$_{Gr}$/sodium acetate = 0.58 g g$^{-1}$.

## 2.3. Analytical method

(1) The concentration of NH$_4^+$-N, NO$_2^-$-N and NO$_3^-$-N and COD were determined according to standard methods [27]; gravimetric methods were used to determine the mixed liquor suspended solids (MLSS) and mixed liquor volatile suspended solids (MLVSS); temperature was monitored by a WTW Multi 3420i meter (WTW Company, Germany). The total inorganic nitrogen (TIN) was defined as the sum concentration of NH$_4^+$-N, NO$_2^-$-N and NO$_3^-$-N.

(2) Using SPSS software to analyse the correlation of experimental parameters. The correlation coefficient $r$ represents the degree of correlation between the two parameters.

**Table 2.** The operation of continuous flow reactor at different stages with sodium acetate as organic matter.

| stage | time (d) | concentration (mg l$^{-1}$) | | |
| --- | --- | --- | --- | --- |
| | | NH$_4^+$-N | NO$_2^-$-N | COD (sodium acetate) |
| A | 1–30 | 50 | 66 | 0 |
| B | 31–60 | 50 | 77 | 20 |
| C | 61–90 | 50 | 77 | 35 |
| D | 91–120 | 50 | 77 | 50 |

(i) When $r > 0$, it means that the two variables are positively correlated, and $r < 0$ means that the two variables are negatively correlated.

(ii) When $|r| \geq 0.8$, it can be considered that the two variables are highly correlated.

(iii) When $0.5 \leq |r| \leq 0.8$, the two variables can be considered to be moderately correlated.

(iv) When $0.3 \leq |r| \leq 0.5$, the two variables can be considered to be low correlation.

(v) When $0 \leq |r| \leq 0.3$, it indicates that the degree of correlation is weak and basically irrelevant.

(3) High-throughput sequencing and microbial population analysis. In this study, high-throughput sequencing technology was used to analyse the bacterial community structure in granular sludge. Bacterial 16S rRNA genes were PCR-amplified with primer pairs, 341F (CCTACGGGNGGCWGCAG) and 805R (GACTGGAGTTCCTTGGCACCC) for the V3 and V4 region. At different stages of the experiment, the granular sludge was taken from the reactor for high-throughput sequencing analysis. We marked the granular sludge as R1 which was taken from the stable operation period (stage 1 and stage A); marked the granular sludge as R2 which was taken from the stage 4; and marked the granular sludge as R3 which was taken from the stage D.

## 2.4. Calculation method

Anammox coupled denitrification process and nitrogen removal performance in the two reactors were analysed separately. The calculation formulae are as follows:

$$\Delta C(\text{NO}_2^-\text{-N})_{\text{anammox}} = \left[ C(\text{NH}_4^+\text{-N})_{\text{in}} - C(\text{NH}_4^+\text{-N})_{\text{eff}} \right] \times 1.32 \tag{2.1}$$

and

$$\Delta C(\text{NO}_2^-\text{-N})_{\text{denitrification}} = \Delta C(\text{NO}_2^-\text{-N})_{\text{total}} - \Delta C(\text{NO}_2^-\text{-N})_{\text{anammox}} \tag{2.2}$$

The proportion of NO$_2^-$-N consumed in the anaerobic ammonia oxidation process ($\eta_1$) and the proportion of NO$_2^-$-N consumed in the denitrification process ($\eta_2$) are calculated as follows:

$$\eta_1 = \frac{\Delta C(\text{NO}_2^-\text{-N})_{\text{anammox}}}{\Delta C(\text{NO}_2^-\text{-N})_{\text{total}}} \tag{2.3}$$

and

$$\eta_2 = 1 - \eta_1, \tag{2.4}$$

Where $\Delta C(\text{NO}_2^-\text{-N})_{\text{denitrification}}$ is NO$_2^-$-N removal via denitrification; $\Delta C(\text{NO}_2^-\text{-N})_{\text{anammox}}$ is NO$_2^-$-N removal via anammox; $\Delta C(\text{NO}_2^-\text{-N})_{\text{total}}$ is total NO$_2^-$-N removal in the reactor; $C(\text{NO}_2^-\text{-N})_{\text{in}}$ and $C(\text{NO}_2^-\text{-N})_{\text{eff}}$ are the NO$_2^-$-N concentration of the influent and effluent, respectively.

# 3. Results and discussion

## 3.1. Comparison of nitrogen removal performance with different organics

### 3.1.1. Nitrogen removal performance when glucose was organic matter

When nitrite was sufficient as the substrate in reactor, the effect of glucose on nitrogen removal performance of the anammox reactor is shown in figure 2. From stage 1 to stage 4, the average

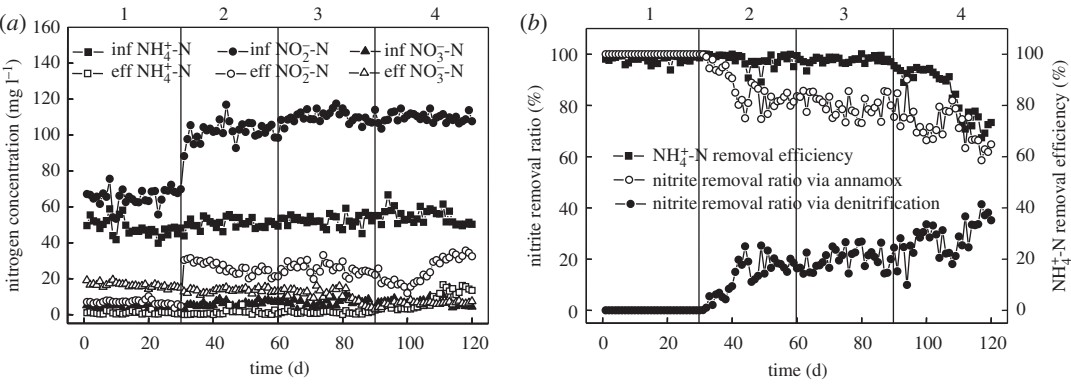

**Figure 2.** Nitrogen removal performance of the reactor with glucose as organic matter. (*a*) Nitrogen removal performance of the reactor. (*b*) The ammonium removal efficiency and nitrite removal ratio via different processes in the reactor.

concentration of $NH_4^+$-N in influent of the reactor was maintained at 50 mg l$^{-1}$ and the HRT was 0.96 h. In stage 1 (0 ~ 30 d), no glucose was added into influent, and the reactor had good anammox performance. The average concentrations of $NH_4^+$-N, $NO_2^-$-N and $NO_3^-$-N in influent were 50, 66 and 4.0 mg l$^{-1}$, respectively, while in effluent the concentrations were 1.2, 7.1 and 15.6 mg l$^{-1}$, respectively. The average removal rates of $NH_4^+$-N and $NO_2^-$-N were 97.5% and 90.0%.

Different concentrations of glucose were added into influent from stage 2 to stage 4, and the COD concentrations were 50, 100 and 200 mg l$^{-1}$, respectively. The average influent $NO_2^-$-N concentrations of stage 2–4 were 100, 110 and 110 mg l$^{-1}$, respectively, to ensure it is sufficient in the reactor.

In stages 2 and 3, $NH_4^+$-N removal were relatively stable. The average $NH_4^+$-N concentration of the effluent and the average $NH_4^+$-N removal efficiency were 1.0 mg l$^{-1}$ and 97.8%, respectively. But in stage 4, the removal of $NH_4^+$-N gradually deteriorated while its concentration increased in effluent, indicating that the anammox activity of the reactor decreased. By the 120th day, the effluent $NH_4^+$-N concentration increased to 13.4 mg l$^{-1}$, and the $NH_4^+$-N removal efficiency decreased to 73.4%. From stage 2 to stage 4, with the increase of influent glucose concentration, the ratio of $NO_2^-$-N consumed via anammox process decreased gradually while the ratio of $NO_2^-$-N consumed via denitrification process increased. The consumption of $NO_2^-$-N in anammox process in stage 4 decreased from 75.5 to 61.9%.

Some researchers have studied the effects of glucose organics on anammox coupled denitrification sludge. When Qin *et al.* [19] increased the concentration of glucose to 374.3 mg l$^{-1}$, the $NH_4^+$-N removal efficiency was reduced to 11.8%; when the concentration of organic matter was reduced to 56.4 mg l$^{-1}$ and the concentration of $NO_2^-$-N was increased simultaneously, $NH_4^+$-N removal efficiency was rapidly increased to 96%, and the anammox activity of sludge was restored to the level before adding organic matter. Chamchoi *et al.* [20] experimented with fat milk as organic matter, and the results showed that when the COD concentration of influent was 100–200 mg l$^{-1}$, AAOB could effectively compete for $NO_2^-$-N; but when the influent COD concentration exceeded 300 mg l$^{-1}$, the activity of AAOB in the reactor was strongly inhibited. Another research by Tang *et al.* [7] showed that when the influent COD concentration was 100, 200 and 300 mg l$^{-1}$ respectively, AAOB had good activity, and the proportion of the $NO_2^-$-N removal via anammox accounted for 83.3%, 65.3% and 55.3%, respectively, of total $NO_2^-$-N removal. When the influent COD concentration was 700 mg l$^{-1}$, nitrogen removal performance of AAOB in the reactor deteriorated, and the ratio of $NO_2^-$-N consumed by anammox process was decreased to 2.1%. However, in both the studies of Chamchoi *et al.* [20] and Tang *et al.* [7], when organics exerted a strong inhibitory effect on anammox performance of the reactor, the substrate $NO_2^-$-N in the reactor was in an insufficient state, which could not well reflect the endurance of the reactor to organics.

In this study, when the concentration of glucose was low, anammox activity remained high under sufficient nitrite condition. But when the concentration of glucose was 200 mg l$^{-1}$, anammox reactor could not effectively endure the organic matter even if there was sufficient influent nitrite nitrogen. The results were similar to those of Chamchoi *et al.* [20] and Tang *et al.* [7]. The reason could be analysed as follows: in the system, denitrifying bacteria and AAOB simultaneously used the substrate (organic matter, $NH_4^+$-N and $NO_2^-$-N) for their own metabolic growth [28,29]. However, denitrifying bacteria is heterotrophic bacteria, whose cell yield rate is much higher than AAOB, so denitrifying bacteria gradually occupied the effective space of granular sludge [30]. Accordingly, the biological density of AAOB in granular sludge decreased, which resulted in the deterioration of anammox activity in the reactor.

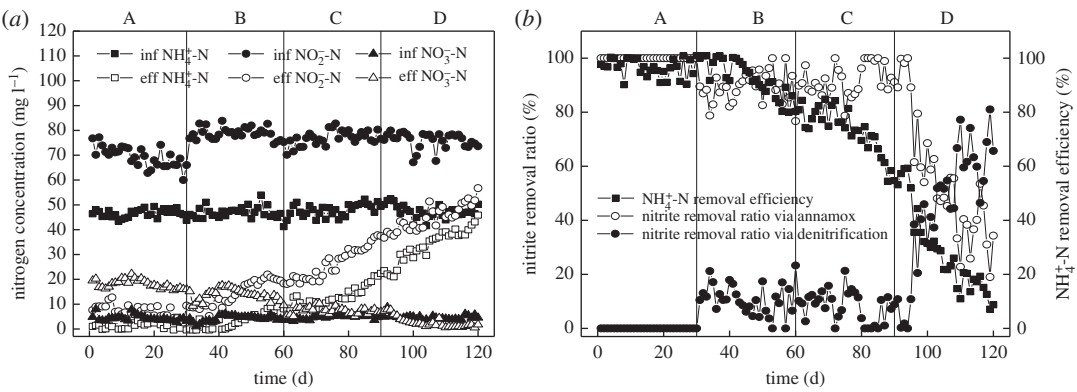

**Figure 3.** Nitrogen removal performance of the reactor with sodium acetate as organic matter. (*a*) Nitrogen removal performance of the reactor. (*b*) The ammonium removal efficiency and nitrite removal ratio via different processes in the reactor.

### 3.1.2. Nitrogen removal performance when sodium acetate was organic matter

When nitrite was sufficient as the substrate, the effect of sodium acetate on nitrogen removal performance of the anammox reactor is shown in figure 3. From stage A to stage D, the average influent $NH_4^+$-N concentration and the HRT were both consistent with that in stage 1 to stage 4. In stage A, no sodium acetate organic matter was added into the influent, and the reactor had good anammox activity. The average concentrations ($NH_4^+$-N, $NO_2^-$-N and $NO_3^-$-N) of influent and effluent were similar with that in stage 1 when glucose was organic. Different concentrations of sodium acetate organics were added into the influent from stage B to stage D with COD concentrations of 20, 35 and 50 mg l$^{-1}$, respectively. The influent $NO_2^-$-N concentration of the reactor from stage B to stage D was 77 mg l$^{-1}$ on average to ensure it is sufficient.

Figure 3 shows that $NH_4^+$-N removal efficiency remained stable in the first 12 days of stage B (the 31st to 42nd days), and it maintained at a relatively high value as 98.14%. However, from the 43rd day, $NH_4^+$-N removal efficiency began to decrease, and in stage D, it fell to 8.74%. The $NO_2^-$-N removal ratio via anammox also decreased significantly from stage B to stage D while it increased via denitrification. In stage B, the $NO_2^-$-N removal ratio via anammox averaged 89.47%. At this point, anammox played a major role in the coupling reaction. In stage D, the $NO_2^-$-N removal ratio via anammox decreased to 34.32%. Anammox lost the dominant state in the coupling system.

When sodium acetate was used as organic and sufficient $NO_2^-$-N was ensured in the system, increased COD concentration had no effect on AAOB in short time (first 12 days in stage B). When COD was 20 mg l$^{-1}$, anammox activity was affected, and $NH_4^+$-N removal efficiency decreased significantly. And when COD concentration was 50 mg l$^{-1}$, anammox activity was severely inhibited and the system had poor nitrogen removal performance. But when glucose was the organic matter, while the COD concentration was 50 mg l$^{-1}$, nitrogen removal performed well in the system and anammox activity was not affected. While the COD concentration increased to 200 mg l$^{-1}$, nitrogen removal began to decrease while anammox activity was significantly inhibited. The removal of $NH_4^+$-N was used to reflect the anammox activity, and it was proved that providing sufficient $NO_2^-$-N did not relieve the inhibition of AAOB by sodium acetate. The reason could be related to the different metabolism of glucose and sodium acetate. Sodium acetate is a small molecule organic matter, which is easier to be metabolized than glucose and has a greater impact on AAOB [31,32]. Zheng *et al.* [33] showed that the removal of $NH_4^+$-N was 31.1 and 15.0 mg l$^{-1}$ when glucose and sodium acetate were used as organics, respectively, under the same C/N. It was obvious that the inhibition of sodium acetate on anammox was stronger. Dapena-Mora *et al.* [4] added 25 and 50 mM sodium acetate to the reactor, and the anammox process was inhibited by 22% and 70%, respectively, indicating that higher sodium acetate concentration had a greater effect on AAOB. In the first 12 days of this experiment, nitrogen removal performance of the reactor was stable and maintained at a high level. On the one hand, the reason could be that the operation time with sodium acetate was short, so the reactor was in adaptation stage and the effect of sodium acetate on anammox was not obvious. On the other hand, it could be related to the diversity of anammox metabolism pathway [34,35].

### 3.2. Pearson correlation analysis

Tables 3 and 4 show the correlation coefficients of various parameters in the system when glucose and sodium acetate were used as organics, respectively. In the table, COD is the initial concentration of organics added into the reactor, and $\Delta C$ $NH_4^+$-N, $\Delta C$ $NO_2^-$-N, $\Delta C$ $NO_3^-$-N, $\Delta C$ TN are $NH_4^+$-N removal,

**Table 3.** Pearson correlation coefficients of various parameters with glucose as organic matter.

| | correlation coefficient $r$ | | | | | |
| --- | --- | --- | --- | --- | --- | --- |
| | COD | $\Delta C\ NH_4^+$-N | $\Delta C\ NO_2^-$-N | $\Delta C\ NO_3^-$-N | $\Delta C$ TN | η1 |
| COD | 1 | −0.080 | 0.727[a] | −0.868[a] | 0.702[a] | −0.846[a] |
| $\Delta C\ NH_4^+$-N | | 1 | 0.399[a] | −0.040 | 0.541[a] | 0.199[a] |
| $\Delta C\ NO_2^-$-N | | | 1 | −0.720[a] | 0.970[a] | −0.784[a] |
| $\Delta C\ NO_3^-$-N | | | | 1 | −0.767[a] | 0.743[a] |
| $\Delta C$ TN | | | | | 1 | −0.674[a] |
| η1 | | | | | | 1 |

[a]Correlation is significant at the 0.01 level (2-tailed).

**Table 4.** Pearson correlation coefficients of various parameters with sodium acetate as organic matter.

| | correlation coefficient $r$ | | | | | |
| --- | --- | --- | --- | --- | --- | --- |
| | COD | $\Delta C\ NH_4^+$-N | $\Delta C\ NO_2^-$-N | $\Delta C\ NO_3^-$-N | $\Delta C$ TN | η1 |
| COD | 1 | −0.827[a] | −0.808[a] | −0.957[a] | −0.756[a] | −0.687[a] |
| $\Delta C\ NH_4^+$-N | | 1 | 0.951[a] | 0.872[a] | 0.978[a] | 0.894[a] |
| $\Delta C\ NO_2^-$-N | | | 1 | 0.872[a] | 0.980[a] | 0.740[a] |
| $\Delta C\ NO_3^-$-N | | | | 1 | 0.812[a] | 0.710[a] |
| $\Delta C$ TN | | | | | 1 | 0.823[a] |
| η1 | | | | | | 1 |

[a]Correlation is significant at the 0.01 level (2-tailed).

$NO_2^-$-N removal, $NO_3^-$-N production and total nitrogen (TN) variation, respectively. η1 represents the proportion of $NO_2^-$-N consumed via anammox process.

When glucose was used as organics, the correlation of $NH_4^+$-N removal and COD was weak. The reason could be that different concentrations of glucose had quite different effects on the reactor, making the correlation irregular. The removal of $NO_2^-$-N and TN were positively correlated with COD, and the production of $NO_3^-$-N was negatively correlated with COD. It was proved that with the increase of COD concentration, the removal amount of $NO_2^-$-N and TN increased, the production amount of $NO_3^-$-N decreased, and denitrification occurred in the system, which was consistent with the description in §3.1.1.

When sodium acetate was used as organics, the $NH_4^+$-N removal and COD showed a significantly negative correlation. And the decrease of $NH_4^+$-N removal proved that sufficient $NO_2^-$-N in the reactor did not relieve the effect of sodium acetate on anammox. The removal of $NO_2^-$-N, the production of $NO_3^-$-N and the removal of TN were negatively correlated with COD. It was proved that the anammox activity was inhibited by sodium acetate. The nitrogen removal performance of the coupling system deteriorated, which was consistent with the description in §3.1.2.

There were both a strong positive correlation between the removal of $NO_2^-$-N and TN when glucose and sodium acetate were used as organics. This was mainly due to the fact that $NO_2^-$-N is the common reaction substrate for anammox and denitrification [6,7]. Also, the ratios of $NO_2^-$-N consumed by anammox process both had a strong negative correlation with COD concentration, which was illustrated that as the concentration of COD increased, the proportion of $NO_2^-$-N removed by denitrification increased in the system and the denitrification activity increased. This was consistent with the studies by Tomar *et al.* [36,37].

## 3.3. The change of appearance morphology of granular sludge

Figure 4 shows the photographs of the granular sludge from the reactor at each stage when glucose and sodium acetate were organic, respectively. The granular sludge in the initial stage of the reaction was

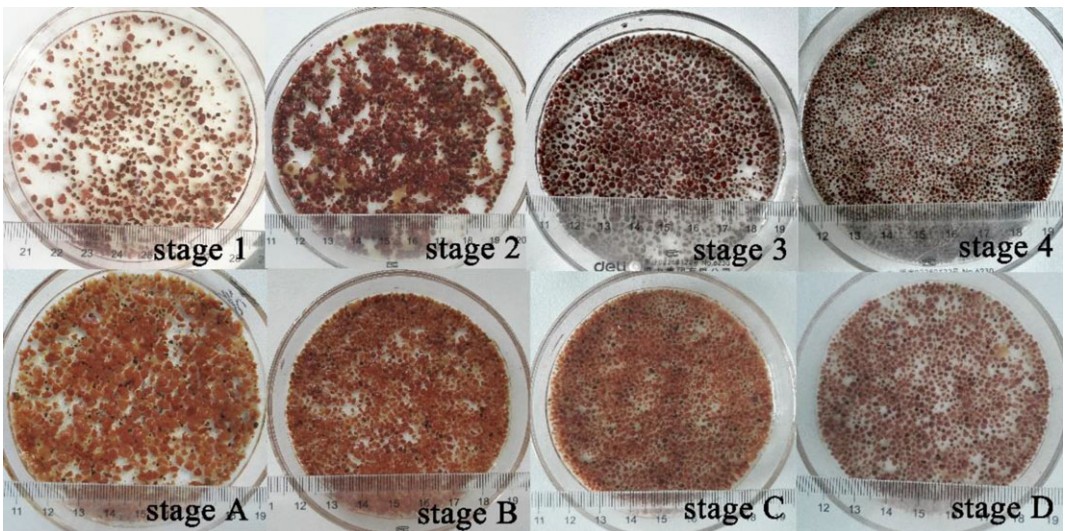

**Figure 4.** The appearance morphology of granules at different stages.

bright brick red. With the increase of the organic concentration, the black area of the granular sludge gradually increased. Some studies showed that the activity of AAOB was directly related to the content of haem C [38–40], which indicated that in this experiment, the biological density of AAOB in the granular sludge decreased gradually and the biological density of denitrifying bacteria increased at the same time. Thereby, the anammox activity of the granular sludge was decreased. The particle size of the granular sludge was also gradually decreased as the concentration of the organic increased [41].

## 3.4. Analysis of microbial population structure of granular sludge

The population change of granular sludge was investigated by high-throughput sequencing, and the effect of organic on AAOB was explored. In the sequencing results, the species diversity index *Coverage* of the three samples reached 1.00, 1.00 and 0.98, respectively, indicating that the sequencing results can represent the real situation of the sample.

Figure 5*a* shows the classification of high-throughput sequencing results for the three samples at the phylum level. *Proteobacteria* was the dominant bacteria for all three samples, and the proportion of *Proteobacteria* in R1 (taken from stage 1 and stage A) was 44.03%. The ratio of it in R2 (taken from stage 4) and R3 (taken from stage D) increased to 67.79% and 65.67%, respectively, after adding organic. *Proteobacteria* contains most of the denitrifying bacteria, so it was obvious that the proportion of denitrifying bacteria increased when organic was added to the system. In R2 and R3, the proportion of *Planctomycetes* and *Acidobacteria* were reduced compared with R1. *Planctomycetes* contains all the genus of known AAOB [42], so the decrease of its abundance was also the main reason for the decrease of the anammox activity in the reactor after the addition of organic [43]. Among the three samples, the proportion of *Bacteroidetes* did not change much, and all three samples had *Chloroflexi* and *Firmicutes*. The sum of abundance of microorganisms in the above six types of phylum accounted for more than 94% in all three samples. Therefore, the microorganisms in granular sludge did not change much at the phylum level after adding different organic in this experiment.

Figure 5*b* shows the sequencing results of the genus level of the samples. It can be seen from the pie plots that the main AAOB of R1, R2 and R3 were all *Candidatus Kuenenia*, accounting for 20.42%, 8.93% and 2.3%, respectively. It was obvious that after adding organic into the reactor the genus of AAOB did not change. The main reason could be that *Candidatus Kuenenia* had a high matrix tolerance [44]. However, with the increasing concentration of organic, the proportion of *Candidatus Kuenenia* in stage 4 and stage D were lower than that in the initial stage (stage 1 and stage A), and *Candidatus Kuenenia* in sodium acetate was 6.63% lower than that in glucose. Therefore, it could be concluded that the effect of sodium acetate on AAOB was greater than that of glucose [33]. In another study, the number of *Candidatus Kuenenia* decreased significantly after adding sodium acetate in anammox reactor [45], which was consistent with the results of this experiment.

The abundance of *Nitrosamonas* and *Opitutus* in R1 were 1.29% and 0.67%, respectively, which were not detected in R2 and R3. The addition of organic eliminated some bacteria and changed the species and

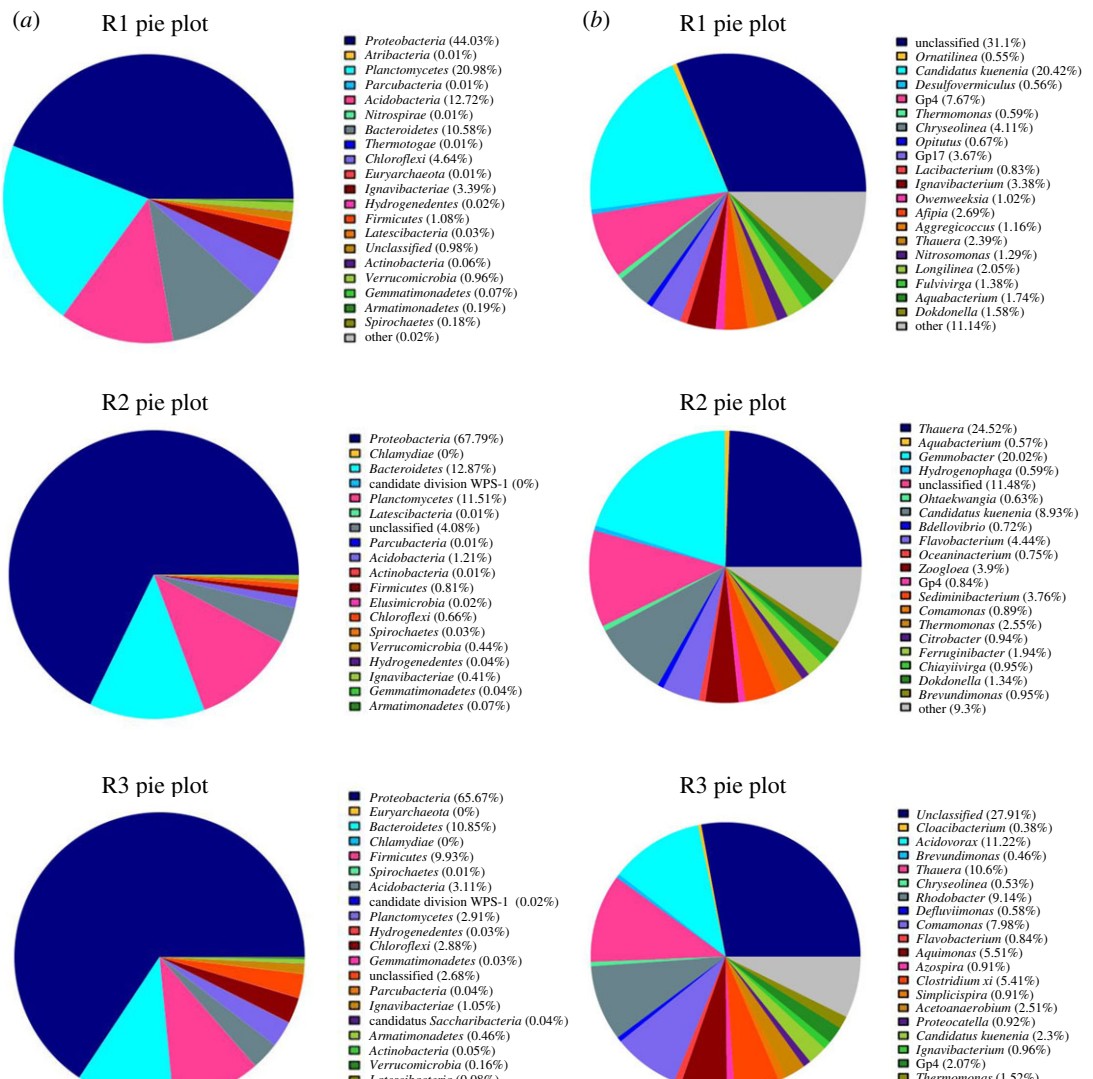

**Figure 5.** The pie plots of the high-throughput sequencing results. (*a*) The high-throughput sequencing results at phylum level. (*b*) The high-throughput sequencing results at genus level.

abundance of bacteria in granular sludge. The denitrifying bacteria in R1 like *Thauera* (2.39%), *Thermomonas* (0.59%), *Aquabacterium* (1.74%), *Longilinea* (2.05%) and *Ignavibacterium* (3.38%) were all with low abundance. The genus of *Thauera, Thermomonas, Flavobacterium, Brevundimonas* and *Comamonas* were in both R2 and R3, which were not detected in R1.

Figure 5*b* shows that the dominant bacteria in R1 were AAOB, and the dominant bacteria in R2 and R3 were denitrifying bacteria. With the addition of organic, the abundance of denitrifying bacteria was higher. AAOB was at a disadvantaged state and it seriously affected the nitrogen removal performance in the system, which was consistent with the changes of nitrogen concentration described above.

## 4. Conclusion

In the UASB reactor with sufficient $NO_2^-$-N, taking glucose as organics, when COD concentration was 50 mg l$^{-1}$, nitrogen removal performance in the system was good, and anammox activity was not affected. When COD concentration increased to 200 mg l$^{-1}$, nitrogen removal performance of the system decreased significantly, and anammox activity was significantly inhibited. But when sodium acetate was used as organic matter, anammox activity began to be affected when COD was 20 mg l$^{-1}$. And when COD concentration was 50 mg l$^{-1}$, anammox was severely inhibited while the system had poor nitrogen removal performance. Adequate nitrite nitrogen could relieve the inhibition of the

coupling system by a low concentration (COD < 200 mg l$^{-1}$) of glucose, but could not relieve the inhibitory effect of sodium acetate. With the increase of organic concentration in the system, the biological density of AAOB in granular sludge gradually decreased, while the biological density of denitrifying bacteria gradually increased. But the dominant AAOB in all three samples is *Candidatus Kuenenia*.

Data accessibility. Data available from the Dryad Digital Repository: https://doi.org/10.5061/dryad.3h1d321 [46].

Authors' contributions. J.Y. carried out the laboratory work, participated in data analysis, participated in the design of the study and drafted the manuscript; J.L. and Z.Z. conceived of the study. L.H., D.L., Y.S. and X.M. coordinated the study. All authors gave final approval for publication.

Competing interests. We have no competing interests.

Funding. Major Science and Technology Program for Water Pollution Control and Treatment (2017ZX07103-001).

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
