## [Reviewer comments · Royal Society Open Science]

Effect of organic matters on anammox coupled denitrification system: when nitrite was sufficient

Jingyue Yang, Jun Li, Zhaoming Zheng, Liangang Hou, Dongbo Liang, Yiqi Sun and
Xiaoran Ma

Article citation details

R. Soc. open sci. **6**: 190771.
<http://dx.doi.org/10.1098/rsos.190771>

Review timeline

Original submission: 16 May 2019
Revised submission: 31 August 2019
Final acceptance: 23 September 2019

Note: Reports are unedited and appear as
submitted by the referee. The review history
appears in chronological order.

Review History

RSOS-190771.R0 (Original submission)

Review form: Reviewer 1

Is the manuscript scientifically sound in its present form?

Yes

Are the interpretations and conclusions justified by the results?

Yes

Is the language acceptable?

No

Is it clear how to access all supporting data?

Not Applicable

Do you have any ethical concerns with this paper?

No

Have you any concerns about statistical analyses in this paper?

No

Recommendation?

Major revision is needed (please make suggestions in comments)

Comments to the Author(s)

The influence of organic matter has been investigated on ANAMMOX by authors. Please pay attention to adding:

1. recent citations
2. Clearly mention research gaps in existing studies.
3. Language should be improved.
4. organic matter may have some inhibitory effects on ANAMMOX discuss how these could be avoided?
5. In you methods, please mention clearly what was your plan to study effects of organic matter on anammox process.

Review form: Reviewer 2

Is the manuscript scientifically sound in its present form?

Yes

Are the interpretations and conclusions justified by the results?

Yes

Is the language acceptable?

Yes

Is it clear how to access all supporting data?

Not Applicable

Do you have any ethical concerns with this paper?

No

Have you any concerns about statistical analyses in this paper?

No

Recommendation?

Accept with minor revision (please list in comments)

Comments to the Author(s)

The author has described the effect of organic matter on the anammox bacteria and its inhibition effect through different forms of carbon sources. The concept seems to be novel and well explored in detail in terms of ammonia removal rate and growth of coupled denitrification system. However, following points should be addressed to enhance the quality of the manuscript:

- 1) The author should described more on how sodium acetate & glucose effect more on anammox bacteria rather their effect on denitrification is more pronounced.
- 2) If author can add biomass yield observed during the course of feeding sodium acetate and glucose, respectively, to arrive at their effect on their specific growth rate.

- 3) In all the experiment hydraulic retention time (HRT) maintained was 0.96h. The author mention the justification for maintaining such a lower retention time.
- 4) References need to be updated.

Review form: Reviewer 3

Is the manuscript scientifically sound in its present form?

Yes

Are the interpretations and conclusions justified by the results?

Yes

Is the language acceptable?

No

Do you have any ethical concerns with this paper?

No

Recommendation?

Major revision is needed (please make suggestions in comments)

Comments to the Author(s)

With the development of water treatment processes, the technology of anammox coupled denitrification has become a research hotspot recently. In the present study, the UASB continuous flow experiment was carried out to investigate the effect of the concentration of glucose and sodium acetate on nitrogen removal performance of anammox reactor under the condition of sufficient nitrite nitrogen. Some novel and interesting results were presented. The manuscript could provide a practical guidance for applying in the process of anammox coupled denitrification. And it is appropriate for Royal Society Open Science. But the following questions need to be addressed before consideration of publication.

- 1) Introduction should be optimized further. More recently published references on the effect of the concentration of organics on nitrogen removal performance of anammox coupled denitrification should be cited and reviewed. And it would be better if the logical structure of introduction is adjusted and perfected.
- 2) The author need to explain why they choose UASB instead of other types of reactors.
- 3) When ensuring nitrite nitrogen is sufficient as a substrate, why did the author choose these nitrite nitrogen concentrations (100/110,77 mg/L)?
- 4) When investigating the effect of the concentration of glucose and sodium acetate on nitrogen removal performance of anammox coupled denitrification, I think the concentration gradient set by the authors is too few.
- 5) The annotation font of Figure 3a and Figure 3b is incorrect. Please modify it and all similar errors in this article should be re-examined and modified according to journal guide.
- 6) Line 276-278 of Page 10, please explain the reason why the biological density of AAOB and denitrifying bacteria has changed, and how to prove it.
- 7) Please explain which reactor and which stage the sludge of R1, R2 and R3 are taken from in "3.4 Analysis of microbial population structure of granular sludge".
- 8) All format of reference citation in the text should be re-examined and modified according to journal guide.

Decision letter (RSOS-190771.R0)

15-Aug-2019

Dear Miss Yang,

The editors assigned to your paper ("Effect of organic matters on anammox coupled denitrification system: when nitrite nitrogen was sufficient") has now received comments from reviewers. We would like you to revise your paper in accordance with the referee and Editor suggestions which can be found below (not including confidential reports to the Editor). Please note this decision does not guarantee eventual acceptance.

Please submit a copy of your revised paper before 06-Sep-2019. Please note that the revision deadline will expire at 00.00am on this date. If we do not hear from you within this time then it will be assumed that the paper has been withdrawn. In exceptional circumstances, extensions may be possible if agreed with the Editorial Office in advance.

Your manuscript will be returned to at least the original reviewers. In the event that any of these individuals are not available to review, additional reviewers will be sought by the editors.

Please note that the the journal does not allow multiple rounds of revision, so we urge you to make every effort to fully address all of the comments at this stage. If you do not fully satisfy the reviewers and editors that you have effectively responded to their critiques, the paper may not be accepted.

- Ethics statement

- Data accessibility

It is a condition of publication that all supporting data are made available either as supplementary information or preferably in a suitable permanent repository. The data accessibility section should state where the article's supporting data can be accessed. This section should also include details, where possible of where to access other relevant research materials such as statistical tools, protocols, software etc can be accessed. If the data has been deposited in an external repository this section should list the database, accession number and link to the DOI

for all data from the article that has been made publicly available. Data sets that have been deposited in an external repository and have a DOI should also be appropriately cited in the manuscript and included in the reference list.

If you wish to submit your supporting data or code to Dryad (<http://datadryad.org/>), or modify your current submission to dryad, please use the following link:
<http://datadryad.org/submit?journalID=RSOS&manu=RSOS-190771>

- **Competing interests**

- **Authors' contributions**

- **Acknowledgements**

- **Funding statement**

Kind regards,
Andrew Dunn
Senior Publishing Editor
Royal Society Open Science
openscience@royalsociety.org

Reviewers' Comments to Author:

Reviewer: 1

Comments to the Author(s)

The influence of organic matter has been investigated on ANAMMOX by authors. Please pay attention to adding:

1. recent citations
2. Clearly mention research gaps in existing studies.
3. Language should be improved.
4. organic matter may have some inhibitory effects on ANAMMOX discuss how these could be avoided?
5. In you methods, please mention clearly what was your plan to study effects of organic matter on anammox process.

Reviewer: 2

Comments to the Author(s)

The author has described the effect of organic matter on the anammox bacteria and its inhibition effect through different forms of carbon sources. The concept seems to be novel and well explored in detail in terms of ammonia removal rate and growth of coupled denitrification system. However, following points should be addressed to enhance the quality of the manuscript:

- 1) The author should described more on how sodium acetate & glucose effect more on anammox bacteria rather their effect on denitrification is more pronounced.
- 2) If author can add biomass yield observed during the course of feeding sodium acetate and glucose, respectively, to arrive at their effect on their specific growth rate.
- 3) In all the experiment hydraulic retention time (HRT) maintained was 0.96h. The author mention the justification for maintaining such a lower retention time.
- 4) References need to be updated.

Reviewer: 3

Comments to the Author(s)

With the development of water treatment processes, the technology of anammox coupled denitrification has become a research hotspot recently. In the present study, the UASB continuous flow experiment was carried out to investigate the effect of the concentration of glucose and sodium acetate on nitrogen removal performance of anammox reactor under the condition of sufficient nitrite nitrogen. Some novel and interesting results were presented. The manuscript could provide a practical guidance for applying in the process of anammox coupled denitrification. And it is appropriate for Royal Society Open Science. But the following questions need to be addressed before consideration of publication.

- 1) Introduction should be optimized further. More recently published references on the effect of the concentration of organics on nitrogen removal performance of anammox coupled denitrification should be cited and reviewed. And it would be better if the logical structure of introduction is adjusted and perfected.
- 2) The author need to explain why they choose UASB instead of other types of reactors.
- 3) When ensuring nitrite nitrogen is sufficient as a substrate, why did the author choose these nitrite nitrogen concentrations (100/110,77 mg/L)?
- 4) When investigating the effect of the concentration of glucose and sodium acetate on nitrogen removal performance of anammox coupled denitrification, I think the concentration gradient set by the authors is too few.
- 5) The annotation font of Figure 3a and Figure 3b is incorrect. Please modify it and all similar errors in this article should be re-examined and modified according to journal guide.
- 6) Line 276-278 of Page 10, please explain the reason why the biological density of AAOB and denitrifying bacteria has changed, and how to prove it.

- 7) Please explain which reactor and which stage the sludge of R1, R2 and R3 are taken from in "3.4 Analysis of microbial population structure of granular sludge".
- 8) All format of reference citation in the text should be re-examined and modified according to journal guide.

Author's Response to Decision Letter for (RSOS-190771.R0)

See Appendix A.

RSOS-190771.R1 (Revision)

Review form: Reviewer 1

Is the manuscript scientifically sound in its present form?

Yes

Are the interpretations and conclusions justified by the results?

Yes

Is the language acceptable?

Yes

Do you have any ethical concerns with this paper?

No

Have you any concerns about statistical analyses in this paper?

No

Recommendation?

Accept as is

Comments to the Author(s)

Not required

Review form: Reviewer 3

Is the manuscript scientifically sound in its present form?

Yes

Are the interpretations and conclusions justified by the results?

Yes

Is the language acceptable?

Yes

Do you have any ethical concerns with this paper?

No

Have you any concerns about statistical analyses in this paper?

No

Recommendation?

Accept as is

Comments to the Author(s)

All comment has already been responded.

Decision letter (RSOS-190771.R1)

23-Sep-2019

Dear Miss Yang,

I am pleased to inform you that your manuscript entitled "Effect of organic matters on anammox coupled denitrification system: when nitrite was sufficient" is now accepted for publication in Royal Society Open Science.

Kind regards,

Andrew Dunn

Reviewer comments to Author:

Reviewer: 1

Comments to the Author(s)

Not required

Reviewer: 3

Comments to the Author(s)

All comment has already been responded.

Appendix A

The reply of "Effect of organic matters on anammox coupled denitrification system: when nitrite was sufficient"

Thank you very much for your email with which you sent us the reviewer's report on our paper. We also wish to take this opportunity to thank the reviewer for his constructive comments and valuable recommendations. We have carefully revised the manuscript according to reviewer's suggestion. Our responses to the comments are listed below:

Reviewer: 1

Comments to the Author(s)

The influence of organic matter has been investigated on ANAMMOX by authors. Please pay attention to adding:

Comment 1: Please pay attention to adding recent citations.

Reply: Thanks for your comments. Recent citations has been added as required. Such as: Reference 11,21,22,23,24,25,43.

Comment 2: Clearly mention research gaps in existing studies.

Reply: Thanks for your comments. Line 54, we insert: "Moreover, in existing studies, there was no systematic comparison on the effects of different organics on anammox coupled denitrification system when nitrite was sufficient."

Comment 3: Language should be improved.

Reply: Thanks for your comments. Language has been improved as required. Such as: We replace "nitrite nitrogen" with "nitrite" in the revised manuscript. And other modifications have been noted in the manuscript.

Comment 4: organic matter may have some inhibitory effects on ANAMMOX discuss how these could be avoided?

Reply: The inhibition of anammox by organic substances is divided into toxicity inhibition and non-toxic inhibition. In this experiment, non-toxic organic substances were used, so there was only non-toxic inhibition. And the mainly non-toxic inhibition was competitive inhibition on substrates, which could be avoided by keeping nitrite sufficient in the experiment. Line 60, we insert "In the reactor, nitrite nitrogen was maintained sufficient as substrate to avoid competitive inhibition."

Comment 5: In you methods, please mention clearly what was your plan to study effects of organic matter on anammox process.

Reply: Thanks for your comments. Line 56, we replace "In this study, the effect of glucose and sodium acetate concentration..." with "In this study, to explore the influence threshold of organic matters on anammox under the condition of sufficient nitrite, and to analyze the competitive inhibition characteristics on substrates of anammox coupled denitrification system, the effect of glucose and sodium acetate concentration..."

Reviewer: 2

Comments to the Author(s)

The author has described the effect of organic matter on the anammox bacteria and its inhibition effect through different forms of carbon sources. The concept seems to be novel and well explored in detail in terms of ammonia removal rate and growth of coupled denitrification system. However, following points should be addressed to enhance the quality of the manuscript:

Comment 1: The author should describe more on how sodium acetate & glucose affect more on anammox bacteria rather than their effect on denitrification is more pronounced.

Reply: This manuscript described the effect of organic matters on the anammox coupled denitrification system, therefore the effects on AAOB and denitrifying bacteria were both described. In the manuscript, Fig. 2b and Fig. 3b systematically described the effect on nitrogen removal performance via anammox. The high-throughput sequencing analysis in 3.4 also focused on the analysis of AAOB. The reasons for denitrification described in this manuscript are as follows: after the addition of organic matter, anammox and denitrification reactions were carried out simultaneously. While organic matter was beneficial to denitrification, it inhibited anammox from the side. We could describe anammox more clearly through described denitrification.

According to reviewers' opinions, some descriptions about denitrification were amended. Line 271 we deleted: “...the denitrification activity was high, and...”; Line 289, we deleted “..., and the denitrification activity was increased.”

Comment 2: If author can add biomass yield observed during the course of feeding sodium acetate and glucose, respectively, to arrive at their effect on their specific growth rate.

Reply: I appreciate this suggestion. But this stage of the experiment was over. I am very sorry that I could not add biomass yield observed during the course of feeding sodium acetate and glucose. I will take your suggestion in future experiments. In addition, in this experiment, high-throughput sequencing of the biomass at the beginning and the end of the experiment could provide a general understanding of the effects of organic matter on biomass changes according to 3.4.

Comment 3: In all the experiment hydraulic retention time (HRT) maintained was 0.96h. The author mention the justification for maintaining such a lower retention time.

Reply: The lower hydraulic retention time could result in a higher load in the reactor, which is more conducive to the efficient treatment of sewage by granular sludge. In addition, the low hydraulic retention time could produce a large hydraulic shearing force, which is beneficial to the formation of granular sludge.

According to reviewers' opinions, Line 89, we replaced “...while the hydraulic retention time (HRT) maintained at 0.96h.” with “...while the hydraulic retention time (HRT) maintained at 0.96h to maintain a higher load and a larger hydraulic shearing force in the reactor.”

Comment 4: References need to be updated.

Reply: Thanks for your comments. Recent citations has been added as required. Such as: Reference 11,21,22,23,24,25,43.

Reviewer: 3

Comments to the Author(s)

With the development of water treatment processes, the technology of anammox coupled denitrification has become a research hotspot recently. In the present study, the UASB continuous flow experiment was carried out to investigate the effect of the concentration of glucose and sodium acetate on nitrogen removal performance of anammox reactor under the condition of sufficient nitrite nitrogen. Some novel and interesting results were presented. The manuscript could provide a practical guidance for applying in the process of anammox coupled denitrification. And it is appropriate for Royal Society Open Science. But the following questions need to be addressed before consideration of publication.

Comment 1: Introduction should be optimized further. More recently published references on the effect of the concentration of organics on nitrogen removal performance of anammox coupled denitrification should be cited and reviewed. And it would be better if the logical structure of introduction is adjusted and perfected.

Reply: Thanks for your comments. References has been updated as required. Such as: Reference 11,21,22,23,24,25,43.

And we has adjusted the logical structure of introduction in the revised manuscript.

Comment 2: The author need to explain why they choose UASB instead of other types of reactors.

Reply: In this experiment, UASB was more favorable to the anammox reaction than other reactors. The main reason is that the long residence time of sludge was beneficial to the complete reaction. And the UASB reactor adopted in this experiment had a relatively large height/diameter, which was conducive to sludge sedimentation and granulation.

According to reviewers' opinions, Line 71, we added: “...which was conducive to sludge sedimentation and granulation.”

Comment 3: When ensuring nitrite nitrogen is sufficient as a substrate, why did the author choose these nitrite nitrogen concentrations (100/110,77 mg/L)?

Reply: At the beginning of the experiment, no organic matter was added, ammonia: nitrite was 1.32:1, and nitrite was 66 mg/L. After adding organic matter, in order to ensure sufficient nitrite, the amount of nitrite to be continuously added is the amount of it that can be theoretically consumed via denitrification used all the added organic matter. They are 100 (glucose) and 77 (sodium acetate), respectively. In the subsequent reaction, since the nitrite in the previous stage reactor could not be completely reacted, a small amount (110 mg/L in stage 3) or none (in stage 4/C/D) of nitrite be continued to add to ensure the sufficient but not too much nitrite in the experiment.

Line 91-94: we replaced “...,the initial concentration of nitrite nitrogen in influent water was the sum of nitrite nitrogen consumed by COD and anammox” with “the initial concentration of nitrite in the influent was the sum of nitrite theoretically consumed by denitrification (assuming that all the added COD was consumed) and anammox”.

Comment 4: When investigating the effect of the concentration of glucose and sodium acetate on nitrogen removal performance of anammox coupled denitrification, I think the concentration gradient set by the authors is too few.

Reply: Thanks for your comments. Referring to the previous experiments, too low concentration difference has little effect on the reaction results, so the concentration difference as glucose 50~100mg/L, sodium acetate 15~20 mg/L was chosen. Indeed, we wanted to carry out more gradient concentration experiments, but as the reaction progressed, the influent concentration became too high which caused irreversible inhibition to the reactor, and the concentration of the effluent was extremely high. So there was no practical significance to continue to increase the concentration gradient. We will pay attention to this in future experiments and we hope you could understand.

Comment 5: The annotation font of Figure 3a and Figure 3b is incorrect. Please modify it and all similar errors in this article should be re-examined and modified according to journal guide.

Reply: The annotation font of Figure 3a and Figure 3b has been correct as required. And other similar errors also has been modified according to journal guide. Line 128-133, and Figure 4-5.

Comment 6: Line 276-278 of Page 10, please explain the reason why the biological density of AAOB and denitrifying bacteria has changed, and how to prove it.

Reply: The addition of organic matter had an effect on both AAOB and denitrifying bacteria. Organic matter was beneficial to the growth of denitrifying bacteria, and the density of denitrifying bacteria was increased. As we all know, organic matter was detrimental to AAOB, and the presence of denitrifying bacteria produces competitive inhibition on AAOB, resulting in a decrease in AAOB density. And we explained it in line 283-288 according to Fig.4. What's more, in this experiment, the change of bacterial density was confirmed by high-throughput sequencing. In 3.4, Fig5a, line301-309, fig5b, line325-327 and 338-342.

Comment 7: Please explain which reactor and which stage the sludge of R1, R2 and R3 are taken from in “3.4 Analysis of microbial population structure of granular sludge”.

Reply: Line 301-304, we replace “...and the proportion of *Proteobacteria* in R1 was 44.03%. The ratio of it in R2 and R3 increased to 67.79% and 65.67% respectively after adding organic.” with “...and the proportion of *Proteobacteria* in R1(taken from stage 1 and stage A) was 44.03%. The ratio of it in R2 (taken from stage 4) and R3 (taken from stage D) increased to 67.79% and 65.67% respectively after adding organic.” In the previous description, we also mentioned it. Line 121-123 in 2.3.

Comment 8: All format of reference citation in the text should be re-examined and modified according to journal guide.

Reply: Thanks for your comments. The format of reference citation has been modified according to journal guide.